# BatchCP: Forecasting Time-Series Data That Have Change Points

## Abstract

Many methods for time-series forecasting are known in classical statistics, such as autoregression, moving averages, and exponential smoothing. Some novel, recent approaches for time-series forecasting based on deep learning have shown very promising results already. However, time series often have change points, which can degrade the prediction performance substantially. This paper extends existing frameworks by detecting and including those change points. We show that our method, called BatchCP, performs as well as standard frameworks when there are no change points and considerably better when there are change points. More generally, we show that the batch size provides an effective and surprisingly simple way to deal with change points in architectures in modern forecasting models, such as DeepAR, Transformers, and TFTs.

## 1 Introduction

Time-series forecasting has evolved from classical statistical methods to sophisticated deep learning architectures, yet the fundamental challenge of handling structural breaks in temporal data remains largely unresolved. Classical time-series analysis, as comprehensively surveyed by De Gooijer and Hyndman (2006), established foundational approaches including autoregressive integrated moving average (ARIMA) models, exponential smoothing, and state space methods. These traditional methods, while effective for stationary or slowly evolving processes, struggle when confronted with abrupt changes in the underlying data-generating mechanism. The advent of deep learning has revolutionized time-series forecasting capabilities. Graves (2013) demonstrated the power of recurrent neural networks for sequence modeling, while Sutskever et al. (2014) established the sequence-to-sequence framework that became fundamental to modern forecasting architectures. Long Short-Term Memory (LSTM) networks, introduced by Hochreiter and Schmidhuber (1997) and refined by Gers et al. (2001), provided the foundation for handling long-term dependencies in temporal data. Building upon these advances, Salinas et al. (2020) introduced DeepAR, which demonstrated how probabilistic forecasting with autoregressive recurrent networks could handle multiple related time series simultaneously while producing uncertainty estimates. The success of attention mechanisms in natural language processing, culminating in the Transformer architecture by Vaswani et al. (2017), sparked significant interest in applying these techniques to time-series forecasting. Zhou et al. (2021) introduced the Informer model specifically designed for long sequence time-series forecasting, addressing the quadratic computational complexity of standard attention mechanisms. Lim et al. Lim et al. (2021) developed Temporal Fusion Transformers (TFTs), which combine the benefits of recurrent and attention-based architectures while incorporating static covariates and known future inputs. Wu et al. Wu et al. (2021) proposed Autoformer, which introduces decomposition capabilities within the Transformer framework to better handle trend and seasonal components in time series. Despite these architectural advances, the problem of change points—abrupt shifts in the statistical properties of time series—remains a significant challenge. Change points are ubiquitous in real-world data: equipment maintenance cycles in industrial IoT systems Zhao et al. (2019), regime changes in financial markets Ang and Bekaert (2002), policy interventions in economic data Bai and Perron (2003b), seasonal transitions in environmental monitoring Reeves et al. (2007), and structural breaks in energy consumption patterns Haben et al. (2021). The presence of change points violates the fundamental stationarity assumptions underlying most forecasting methods, leading to degraded prediction performance and unreli-

able uncertainty estimates. The change point detection literature has developed sophisticated methods for identifying structural breaks in time series. Classical approaches include the CUSUM test (Page (1954)) and its variants, which detect changes in mean or variance. Killick et al. (2012) developed optimal partitioning methods for multiple change point detection, while Fearnhead (2006) introduced efficient exact algorithms for this problem. The MOSUM (Moving Sum) approach, as implemented by Meier et al. (2021), provides robust detection of mean changes in time series. More recent work by Li et al. (2022) explores automatic change point detection using deep learning techniques, while Yamin et al. (2022) focuses on online detection methods suitable for streaming applications. However, the integration of change point detection with modern forecasting methods needs to be explored further. Traditional approaches to handling change points in forecasting fall into several categories, each with significant limitations. Data preprocessing methods, as discussed in the comprehensive survey by Aminikhanghahi and Cook (2017), typically remove change points or replace them with interpolated values. This approach, while simple, distorts the natural structure of the time series and discards potentially valuable information about the underlying process dynamics. Segmentation approaches, advocated by some practitioners, split time series at detected change points and treat each segment independently. While this preserves the stationarity within segments, it loses important temporal relationships across change points and significantly reduces the amount of available training data for each model. A particularly problematic approach, yet unfortunately common in practice, is simply ignoring change points entirely. When modern deep learning forecasting methods encounter change points during training, they attempt to learn these discontinuities as part of the normal temporal pattern. This leads to several critical issues: models may predict change points where none exist, uncertainty estimates become unreliable around historical change points, and the learned representations fail to capture the true underlying dynamics of the process. The batch-based training procedures used by virtually all modern forecasting architectures exacerbate this problem, as randomly sampled training windows frequently span change points, creating inconsistent learning signals. Recent work has begun to address some aspects of this challenge. Rangapuram et al. (2018) explored state space models for handling non-stationary time series, while Sen et al. (2019) investigated the robustness of various forecasting methods to distributional shifts. However, these approaches either require significant architectural modifications or focus on gradual shifts rather than abrupt change points. The fundamental challenge of training deep learning models on time series with structural breaks remains largely unaddressed in the current literature. The theoretical understanding of how neural networks learn temporal patterns provides important insights into why change points are problematic. Bengio et al. (1994) established that learning long-term dependencies in sequential data is inherently difficult due to the vanishing gradient problem. While LSTM and Transformer architectures have largely solved this technical issue, they rely on the assumption that patterns observed in training data will generalize to future sequences. Change points violate this assumption by introducing fundamental shifts in the data-generating process that cannot be extrapolated from historical patterns alone. Furthermore, the statistical properties of time series with change points present unique challenges for modern forecasting methods. As demonstrated by Bai and Perron Bai and Perron (2003a), the presence of structural breaks affects not only point forecasts but also prediction intervals and model selection criteria. The probabilistic forecasting capabilities of modern methods like DeepAR, while valuable for quantifying uncertainty, become unreliable when trained on data that spans change points, as the learned probability distributions fail to capture the true uncertainty in the presence of potential future structural breaks.

We propose BatchCP, a fundamentally new approach that addresses the change point problem by working with the natural structure of modern deep learning forecasting methods rather than against it. Our key insight is that virtually all state-of-the-art deep learning forecasting methods learn through batch-wise training procedures, where multiple windowed sequences are randomly sampled from the available time series data. We can leverage this training structure to handle change points without requiring any modifications to the underlying model architectures or loss functions. Instead of manipulating the data or altering model architectures, BatchCP operates at the training batch level, systematically ensuring that no training batch spans a change point while preserving the complete dataset. This approach respects the natural segmentation of the data implied by change points while maintaining the temporal relationships within each stable segment. By training only on temporally consistent data segments, models can learn clean temporal patterns without the confusion that arises from training on sequences that span structural breaks. The core innovation of BatchCP lies in recognizing that change points represent natural boundaries between distinct temporal

regimes, and that respecting these boundaries during training leads to more robust and generalizable models. Unlike previous approaches that view change points as obstacles to be eliminated or worked around, BatchCP treats them as valuable structural information that can be leveraged to improve learning. By intelligently selecting training batches that respect change point boundaries, we enable forecasting models to learn the true underlying dynamics within each regime while avoiding the spurious patterns that arise from training across regime transitions. Our contributions represent significant advances in several key areas. First, we introduce the first general framework for handling change points that works universally across modern deep learning forecasting architectures, including but not limited to DeepAR, Transformers, and TFTs, without requiring any architectural modifications or specialized implementations. Second, we provide a principled theoretical foundation for automatic batch size selection based on change point distribution that maximizes training effectiveness while ensuring computational efficiency. Third, we demonstrate through comprehensive empirical evaluation that our method provides consistent improvements across diverse real-world datasets and multiple forecasting architectures, establishing BatchCP as a universally beneficial enhancement for modern forecasting applications.

## 2 The BatchCP Framework

The BatchCP framework is built on a simple yet powerful principle: change points disrupt local temporal patterns, but preserving data segments between change points maintains valuable predictive information. Consider a time series $\{z_t\}_{t=1}^{T}$ with change points at locations $C = \{c_1, c_2, \ldots, c_k\}$ where $1 < c_1 < c_2 < \ldots < c_k < T$. These change points partition the time series into $k + 1$ segments:

$$S_0 = [1, c_1 - 1], S_1 = [c_1, c_2 - 1], \ldots, S_k = [c_k, T]. \tag{1}$$

Within each segment $S_i$, the data follows a consistent generating process $\mathcal{P}_i$, but the processes differ across segments: $\mathcal{P}_i \neq \mathcal{P}_j$ for $i \neq j$. In standard deep learning forecasting approaches, training batches are created by randomly sampling windows $[t_{start}, t_{start} + L - 1]$ from the entire time series, where $L$ is the fixed window length. The probability that a randomly sampled window spans a change point is significant, especially when change points are frequent. Formally, a window starting at position $t$ spans a change point if there exists some $c_j \in C$ such that $t \leq c_j \leq t + L - 1$. When this occurs, the window contains data from two different generating processes, creating inconsistent training signals that confuse the learning algorithm.

BatchCP fundamentally alters this sampling process by introducing a validity constraint. We define a valid training window as one that lies entirely within a single segment. A window $[t_{start}, t_{start} + L - 1]$ is valid if and only if there exists some segment $S_i$ such that $[t_{start}, t_{start} + L - 1] \subseteq S_i$. This ensures that all data within each training batch comes from the same underlying generating process, eliminating the conflicting signals that arise when training windows span change points. The modified sampling procedure can be formalized as follows. Let $\mathcal{W}_{valid}$ denote the set of all valid window starting positions:

$$\mathcal{W}_{valid} = \{t : \exists i \text{ such that } [t, t + L - 1] \subseteq S_i\} \tag{2}$$

During training, BatchCP samples uniformly from $\mathcal{W}_{valid}$ instead of from the entire range $[1, T - L + 1]$. This modification preserves the stochastic nature of the training process while ensuring temporal consistency within each batch. The choice of window length $L$ becomes critical in this framework. If $L$ is too large relative to the shortest segment, some segments may not contribute any valid windows to the training set. To address this, we introduce an automatic batch size selection algorithm that computes the maximum viable window length as:

$$L_{max} = \left\lfloor \frac{\min_{i=1,\ldots,k} |S_i|}{2} \right\rfloor \tag{3}$$

where $|S_i|$ denotes the length of segment $S_i$.

Our method operates through three key components: change point detection, intelligent batch selection, and automatic batch size optimization. Change point detection in BatchCP can be accomplished through multiple approaches depending on the application context. When change points are known from domain

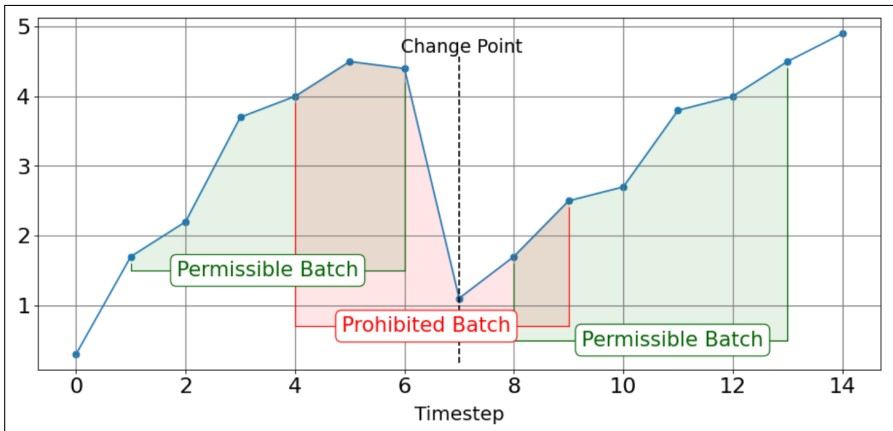

Figure 1: Visualization of the change point method *BatchCP*. It shows an example of a batch selection around a change point. The blue points represent time-series data. The first green area shows a batch created by choosing $t = 1$ as the start index, and with a selected batch size of $s = 6$, the batch contains the points $t = 1$ to $t = 6$. The change point is located at $t = 7$ and is therefore outside the detected batch. The batch is therefore permissible. The red area next to it shows a prohibited batch. The start index of this batch is $t = 4$ and the end index at $t = 9$, so the change point is located inside the batch and is not allowed in training. A new batch must be found. The green area to the right indicates another permissible batch, since the change point lies outside of it.

knowledge, such as scheduled maintenance events or policy changes, they can be specified manually. For cases where change points must be discovered from data, we employ automated detection algorithms such as MOSUM (Moving Sum) statistics, CUSUM, or more sophisticated modern methods. The framework is agnostic to the specific detection method used, making it adaptable to various domains and requirements. Given identified change points $C = \{c_0, c_1, \ldots, c_{k-1}\}$, BatchCP modifies the batch generation process to ensure no batch $[i_{\text{start}}, i_{\text{end}}]$ contains any change point $c_j$. This is achieved through our intelligent batch selection algorithm, which samples batch starting positions uniformly at random but rejects any batch that would span a change point. The process continues until a valid batch is found, ensuring that all training examples come from temporally consistent segments of the data. A critical component of our framework is the automatic determination of optimal batch size. We introduce Algorithm 1 for this purpose, which calculates the maximum allowable batch size based on the distribution of change points in the data.

---

**Algorithm 1** Optimal Batch Size Selection

---

**Require:** Training data $d$, change point detection method $A$
**Ensure:** Maximal batch size $L_{max}$, change points $c_1, \ldots, c_k$
  1: $c_1, \ldots, c_k \leftarrow A(d)$ {Detect change points}
  2: Define segments: $S_0 = [1, c_1 - 1], S_1 = [c_1, c_2 - 1], \ldots, S_k = [c_k, T]$
  3: $L_{max} \leftarrow \lfloor \min_{i=0,\ldots,k} |S_i|/2 \rfloor$
  4: $c_1, \ldots, c_k, L_{max}$

---

The factor of $1/2$ in the batch size calculation ensures that even in the worst case, we can find valid batches between any two consecutive change points. This provides sufficient diversity in training examples while guaranteeing that the batch selection process remains efficient. The algorithm ensures completeness by allowing all data segments between change points to participate in training, thereby maximizing information utilization.

---

**Algorithm 2** Valid Batch Generation

---

**Require:** Training data $d$, batch size $L \leq L_{max}$, segments $S_0, \ldots, S_k$
**Ensure:** Start and end points of valid batch
 1: $n \leftarrow \text{len}(d)$
 2: Compute $\mathcal{W}_{valid} = \{t : \exists i \text{ such that } [t, t + L - 1] \subseteq S_i\}$
 3: **repeat**
 4:     $t_{start} \leftarrow$ uniform random from $\mathcal{W}_{valid}$
 5: **until** valid window found
 6: $t_{end} \leftarrow t_{start} + L - 1$
 7: $t_{start}, t_{end}$

---

The theoretical properties of BatchCP provide important guarantees about the training process. The framework ensures completeness by guaranteeing that all data segments between change points have the opportunity to participate in training, maximizing information utilization. Specifically, for any segment $S_i$ with length $|S_i| \geq L$, there are $|S_i| - L + 1$ valid starting positions within that segment, ensuring adequate representation in the training set. The expected number of sampling attempts to find a valid batch depends on the density of change points and the chosen window length. Let

$$\rho = \frac{|\mathcal{W}_{valid}|}{T - L + 1} \tag{4}$$

denote the fraction of valid starting positions. The expected number of attempts follows a geometric distribution with success probability $\rho$, yielding an expected value of $1/\rho$. Our batch size selection algorithm ensures $\rho$ remains sufficiently large to maintain computational efficiency. Most importantly, the framework makes no assumptions about the underlying forecasting architecture. The modification occurs entirely at the data loading level, making BatchCP universally applicable to any batch-based training procedure. Whether the underlying model is an LSTM-based approach like DeepAR, a Transformer-based architecture, or any other neural forecasting method, the same batch selection principles apply. The mathematical formulation reveals why BatchCP is effective: by ensuring training consistency within the constraint $[t_{start}, t_{start} + L - 1] \subseteq S_i$, we eliminate the distributional mismatch that occurs when training windows span multiple generating processes. This leads to more stable gradients during training and better generalization to unseen data from similar generating processes.

## 3 Universal Application to Forecasting Architectures

To demonstrate the universal applicability of BatchCP, we show how it enhances several state-of-the-art forecasting methods without requiring any architectural modifications. The key insight is that mostly all modern deep learning forecasting methods rely on batch-based training, making our framework immediately applicable. DeepAR represents a prominent example of probabilistic time-series forecasting using LSTM-based architectures. The model learns autoregressive relationships through a conditional distribution

$$Q_{\Theta}(z_{i,t_0:T}|z_{i,1:t_0-1}, x_{i,1:T}) = \prod_{t=t_0}^{T} \ell(z_{i,t}|\theta(h_{i,t}, \Theta)), \tag{5}$$

where $h_{i,t} = h(h_{i,t-1}, z_{i,t-1}, x_{i,t}, \Theta)$ represents the LSTM hidden state. DeepAR's training process naturally creates multiple windows from each time series for training, sampling different starting points to ensure coverage of the prediction range. BatchCP integrates seamlessly by replacing the random window selection with Algorithm 2, ensuring no window spans a change point while maintaining all other aspects of the DeepAR training procedure. Transformer-based forecasting models use attention mechanisms to capture long-range dependencies in time series data. These models have shown remarkable success in various forecasting tasks due to their ability to model complex temporal patterns. BatchCP enhances Transformer-based forecasting by ensuring that the attention mechanisms focus on coherent temporal patterns rather than being confused by change point discontinuities. The integration requires no modification to the attention mechanism itself

- only the batch generation process is altered according to our framework. Temporal Fusion Transformers (TFTs) combine the benefits of LSTM and Transformer architectures with sophisticated attention mechanisms specifically designed for time series forecasting. TFTs incorporate static covariates, known future inputs, and complex temporal patterns through multi-head attention. BatchCP enhances TFTs by ensuring that these sophisticated mechanisms operate on temporally consistent data segments, leading to more effective learning of the underlying patterns. The universality of BatchCP stems from its operation at the data loading level rather than the model level. Since virtually all modern forecasting architectures train using batches of windowed time series data, our framework can be applied as a drop-in replacement for standard batch generation procedures. This design choice makes BatchCP immediately applicable to existing codebases and future forecasting architectures without requiring specialized implementations.

## 4 Experimental Validation

We evaluate BatchCP across four scenarios for each dataset to comprehensively assess its performance. Scenario I uses a naive forecasting method as a baseline, where the current known value is predicted to remain constant in the future. Scenario II applies the standard forecasting method while ignoring change points entirely. Scenario III implements our BatchCP method with manually identified change points based on domain knowledge. Scenario IV uses BatchCP with automatically detected change points using the MOSUM algorithm. This experimental design allows us to assess both the potential of our approach under ideal conditions and its practical applicability with automated change point detection.

### 4.1 Pollution Dataset

Our industrial pollution dataset comes from a recycling company that uses our pipeline in daily operations. The dataset contains 10200 hourly measurements of hydraulic oil pollution in a shredding machine, with 9 known change points corresponding to filter replacement events. These change points occur at irregular intervals due to variable machine usage and deterioration patterns.

We start with Scenario (I), the naive method. It takes the current known value and predicts that it will be the same in the future. The errors for test and training samples are quite high with this method.

In the next step, we ignore the change points in the data and run the vanilla DeepAR model. The batch size is calculated with Algorithm 1. Since the minimum difference is between the change points $c_1 = 1970$ and $c_2 = 2350$, we get the difference $\text{diff}_{1,2} = 380$. Thus, the batch size should be at most

$$L_{\max} = \left\lceil \frac{380}{2} \right\rceil = 190\,.$$

We set $L = 30$. The loss is the Gaussian Likelihood function with parameters $\mu$ and $\sigma$. The model consists of 3 layers: a LSTM layer with 4 units, a dense layer with 3 units, and a Gaussian layer. ReLU is used as the activation function. We do not attempt to optimize the hyperparameters. We split the data set into 60% training samples, 20% validation samples, and 20% test samples.

For vanilla DeepAR (Scenario II), we get a root mean squared error (RMSE) of 98.51 on the training data and 147.83 on the test data. The change points were trained 265 times in the training process.

In Scenario III, we hand-picked the change points of the pollution time series. We know when oil-filter damage occurred to the machine and when the filter was replaced. We picked change points at the following time indices: 700, 1970, 2350, 2730, 3500, 4320, 5050, 6250, 7100. We pass this list of change points in the training process. When the method that generates the random batches is called, we check whether a change point from our list is in the generated batch. If this is the case, we search for a new valid batch. For our method in Scenario III, we get an RMSE of 81.27 on the training data and 112.67 on the test data. Compared to vanilla DeepAR, both the training and test score have improved significantly.

Now, we include the well-known change-point-detection method MOSUM, which consists of procedures for the multiple mean change problem using moving sum statistics. We choose a bandwidth of 0.2, and a value for $\eta = 0.1$ and let the MOSUM method detect the change points of the pollution time series. This can be

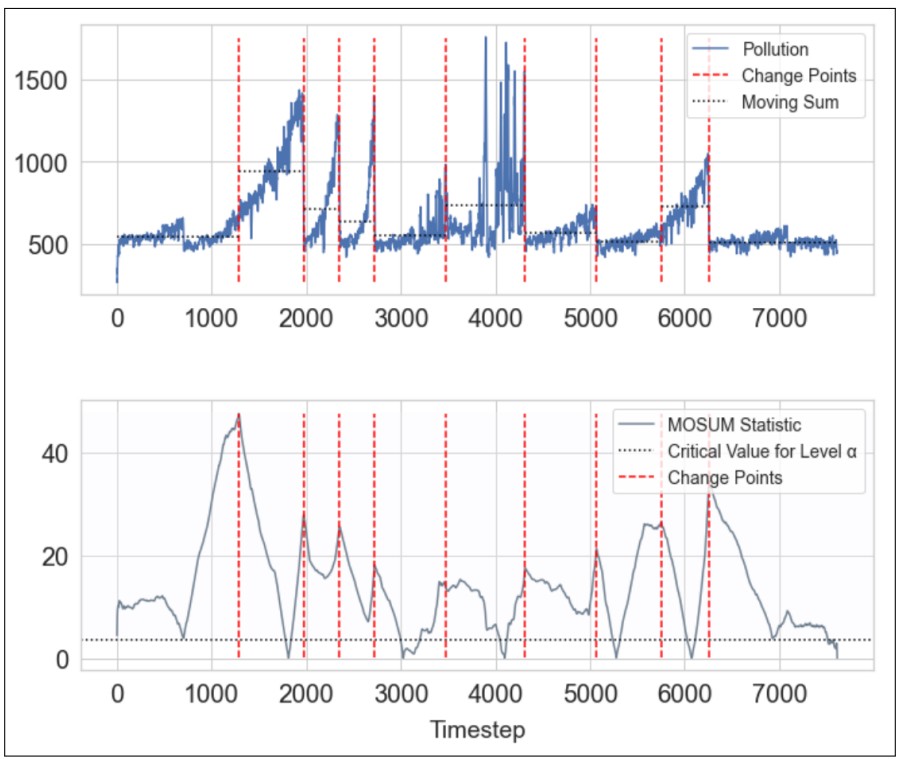

Figure 2: Upper diagram: The pollution time series (blue) in millibar with moving sum (gray) and change points (red) detected with MOSUM. Lower diagram: The MOSUM statistic, which identifies the changes in the mean, the threshold level (dotted line), and the corresponding change points (red). Most of the detected change points equal the actual change points.

Table 1: Prediction error (RMSE) of the different methods on the pollution data. Reported are test and training RMSE of (I) naïve, (II) DeepAR, and our BatchCP equipped with (III) manual and (IV) MOSUM change-point detection.

| SCENARIO | TRAIN | TEST |
|---|---|---|
| Baseline Naive (I) | 198.55 | 282.08 |
| DeepAR-No CPD (II) | 98.51 | 147.83 |
| BatchCP-Manual (III) | 81.27 | 112.67 |
| BatchCP-MOSUM (IV) | 83.42 | 117.75 |

seen in Figure 2. The resulting list of change points is given to the BatchCP model and the further procedure is analogous to the previous experiment. For this method (Scenario IV), we get a training error of 83.42 and a test error of 117.75. The error has worsened slightly, which can be explained by the fact that the MOSUM change point detection is not 100% correct but also missed some points. The error is still smaller than with the DeepAR without taking the change points into account.

The results can be seen in Table 1. Our BatchCP method (Scenario III) is more than three times better than the naïve baseline model (Scenario I) and improves the vanilla DeepAR (Scenario II) by 23.78%. Even with

Table 2: Prediction error (RMSE) of the different methods on the football data. Reported are test and training RMSE of (I) naive, (II) DeepAR, and our BatchCP equipped with (III) manual and (IV) MOSUM change-point detection.

| SCENARIO | TRAIN | TEST |
|---|---|---|
| Baseline Naive (I) | 25.29 | 51.78 |
| DeepAR-No CPD (II) | 15.41 | 25.62 |
| BatchCP-Manual (III) | 8.20 | 14.89 |
| BatchCP-MOSUM (IV) | 10.44 | 18.63 |

Table 3: Prediction error (RMSE) of the different methods on the electricity data. Reported are test and training RMSE of (I) naive, (II) DeepAR, and our BatchCP equipped with (III) manual and (IV) MOSUM change-point detection.

| SCENARIO | TRAIN | TEST |
|---|---|---|
| Baseline Naive (I) | 3.97 | 4.71 |
| DeepAR-No CPD (II) | 1.08 | 1.29 |
| BatchCP-Manual (III) | 0.96 | 1.03 |
| BatchCP-MOSUM (IV) | 1.12 | 1.30 |

automatic change point detection (Scenario IV), which could be further optimized, we are 20.35% better than the vanilla DeepAR.

### 4.2 Football Dataset

The football dataset consists of goal differences for teams in the German Bundesliga, with change points occurring due to season resets and significant roster changes. This dataset presents a different challenge, as the change points are more predictable but still represent genuine shifts in the underlying performance patterns. The dataset consists of dates (the match days) and goal differences for different teams. There are change points because the statistics are reset annually.

The change points are at time indices 31, 65, 99, 133, 157, 174, 191, 208. We calculate the batch size with Algorithm 1 again. The minimum difference is between the change points $c_4$ and $c_5$, so that we get the difference $\text{diff}_{4,5} = 17$. This motivates our batch size

$$L_{\max} = \left\lceil \frac{17}{2} \right\rceil = 9 \,.$$

The batch size is comparably small because there are comparably few observations.

The vanilla DeepAR achieves an RMSE of 25.62, while BatchCP with manual change points reduces this to 14.89, a remarkable 41.9% improvement. The substantial improvement demonstrates that BatchCP is effective across different types of time series and change point patterns. The results can be seen in Table 2.

### 4.3 Electricity Dataset

Our electricity consumption dataset describes the usage patterns of 370 customers over time, with change points representing shifts in consumption behavior due to various factors such as seasonal changes, lifestyle modifications, or economic conditions. The results can be seen in Table 3.

BatchCP with manual change points achieves an RMSE of 1.03 compared to 1.29 for vanilla DeepAR, representing a 20.2% improvement. Interestingly, this dataset shows that MOSUM struggles to identify

Table 4: Prediction error (RMSE) of the different methods on the synthetic data. Reported are test and training RMSE of (I) naive, (II) DeepAR, and our BatchCP equipped with (III) manual and (IV) MOSUM change-point detection.

| SCENARIO | TRAIN | TEST |
|---|---|---|
| Baseline Naive (I) | 3 3.57 | 9.33 |
| DeepAR-No CPD (II) | 2.81 | 5.77 |
| BatchCP-Manual (III) | 2.42 | 5.40 |
| BatchCP-MOSUM (IV) | 2.51 | 5.62 |

Table 5: Prediction errors on the pollution data using Transformer architecture. Reported are test and training RMSE of (I) a naive baseline, (II) Transformer without considering change points and Transformer equipped with (III) manual, and (IV) MOSUM change-point detection.

| SCENARIO | TRAIN | TEST |
|---|---|---|
| Baseline Naive (I) | 198.55 | 282.08 |
| Transformer-No CPD (II) | 75.26 | 96.23 |
| Transformer-Manual (III) | 67.85 | 91.02 |
| Transformer-MOSUM (IV) | 70.22 | 93.12 |

optimal change points, highlighting the importance of domain knowledge in change point detection, though even the MOSUM-based approach does not fall behind the original method.

### 4.4 Synthetic Data Validation

To validate our theoretical predictions under controlled conditions, we generated synthetic data with 3,000 samples and 13 change points. This allows us to assess BatchCP's performance when the ground truth is known. Algorithm 1 allows for a batch size of $L = 50$.

The results confirm our theoretical expectations, with BatchCP showing a 6.4% improvement over vanilla DeepAR even in this controlled setting. This can be seen in Table 4.

### 4.5 Universal Application

We demonstrate BatchCP's universal applicability by testing with multiple architectures. Using Transformer-based forecasting on the pollution dataset, we implement Transformer via Gluon Time Series. The batches are generated according to Algorithm 2.

Vanilla Transformer achieves 96.23 RMSE while BatchCP-enhanced Transformer reduces this to 91.02, a 5.4% improvement (See Table 5). The results show that the Transformer model works better than the DeepAR model in this example, and importantly, we can improve the Transformer noticeably by our BatchCP method for including change points.

### 4.6 Multivariate Time Series

For multivariate time series, we tested with DeepVAR, considering the change points across all dimensions and including them in each time series. In other words, for all time series, only the batches without any change point are used for training. Taking into account the mutual dependencies and cross-correlations between different variables, DeepVAR has proved particularly useful when the relationships between variables are relevant.

Table 6: Prediction errors on the pollution data using multivariate approach. Reported are test and training RMSE of (I) a multivariate baseline, (II) DeepVAR without considering change points and DeepVAR equipped with (III) manual, and (IV) MOSUM change-point detection.

| SCENARIO | TRAIN | TEST |
|---|---|---|
| Baseline Dense (I) | 155.39 | 164.51 |
| Transformer-No CPD (II) | 135.60 | 154.74 |
| Transformer-Manual (III) | 109.76 | 121.69 |
| Transformer-MOSUM (IV) | 118.87 | 126.86 |

Vanilla DeepVAR achieves 154.74 RMSE and BatchCP-DeepVAR reduces this to 121.69, representing an improvement of 21.4%. These results confirm that BatchCP provides consistent benefits across different architectural approaches and demonstrates that our method performs very well also in multivariate settings. The results can be seen in Table 6.

All our results are stable with respect to the randomness (batching, initialization) of the deep-learning pipelines, confirming that BatchCP represents a robust enhancement for modern forecasting applications.

## 5 Analysis and Discussion

BatchCP succeeds because it addresses a fundamental mismatch between the assumptions of modern forecasting methods and the reality of change point data. Forecasting models assume local temporal consistency - that patterns observed in recent history will continue into the near future. Change points violate this assumption by introducing abrupt shifts that confuse learning algorithms. By ensuring that training batches respect change point boundaries, BatchCP allows models to learn cleaner temporal patterns from consistent data segments, leading to better generalization when applied to new data. The computational overhead introduced by BatchCP is minimal. Change point detection represents a one-time preprocessing cost that is typically small compared to the model training time. The batch selection process adds negligible computational burden, as the expected number of sampling attempts remains low even with dense change points due to our principled batch size selection. Once valid batches are generated, the training procedure is identical to the original method, meaning no additional computational cost during the core learning phase. While BatchCP demonstrates consistent improvements across diverse datasets and architectures, several limitations should be acknowledged. The method requires sufficient data between change points for effective batch selection - if change points are extremely dense relative to the minimum required batch size, the method may struggle to find valid training examples. Additionally, the performance of BatchCP with automated change point detection depends on the quality of the detection algorithm used, which may vary across different types of time series. Finally, while our current batch size optimization strategy is effective, further refinement could potentially yield additional improvements. Future research directions include developing change point detection methods specifically optimized for forecasting applications, as current methods are designed for general change point detection rather than forecasting-specific requirements. Theoretical analysis of optimal batch size selection under different change point patterns could provide additional insights for improving the framework. Extension to streaming or online forecasting scenarios, where change points must be detected and handled in real-time, represents another important direction for future work.

## 6 Conclusion

We have introduced BatchCP, a novel framework that fundamentally changes how modern deep learning forecasting methods handle change points. Rather than viewing change points as obstacles to be removed or worked around, BatchCP leverages the natural batch-based structure of modern forecasting methods to systematically exclude problematic training examples while preserving all valuable data.

Our key contributions represent a significant advance in making modern forecasting methods robust to real-world time series characteristics. We have demonstrated a paradigm shift from data manipulation to intelligent training batch selection, provided a universal framework that enhances any batch-based forecasting architecture, established a theoretical foundation for principled batch size optimization, and validated the approach through empirical evaluation showing consistent improvements across diverse real-world applications. BatchCP represents a significant step forward in making modern forecasting methods robust to the realities of real-world time-series data, where change points are not anomalies to be eliminated but fundamental characteristics to be properly handled. The framework's simplicity, generality, and effectiveness make it a valuable addition to the time-series forecasting toolkit. Our results demonstrate that BatchCP can achieve substantial improvements when change points are present while maintaining equivalent performance when they are absent, making it a universally beneficial enhancement for modern forecasting applications.

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

## A    Appendix

We provide supplementary material to the experiments from Section 4.

## B    Synthetic data

We substantiate the empirical results further by considering synthetic data, where we know and control the data-generating process. The data comprises 3000 samples with 13 change points—see Figure 3. The naïve method (I) yields 3.57 RMSE for the training samples and 9.33 RMSE for the test samples.

Algorithm 1 allows for a batch size of $s = 50$. The results for Scenarios (I), (II), (III), and (IV) are in Table 7. BatchCP, both with manually and automatically selected change points, outmatches vanilla DeepAR and naïve once more.

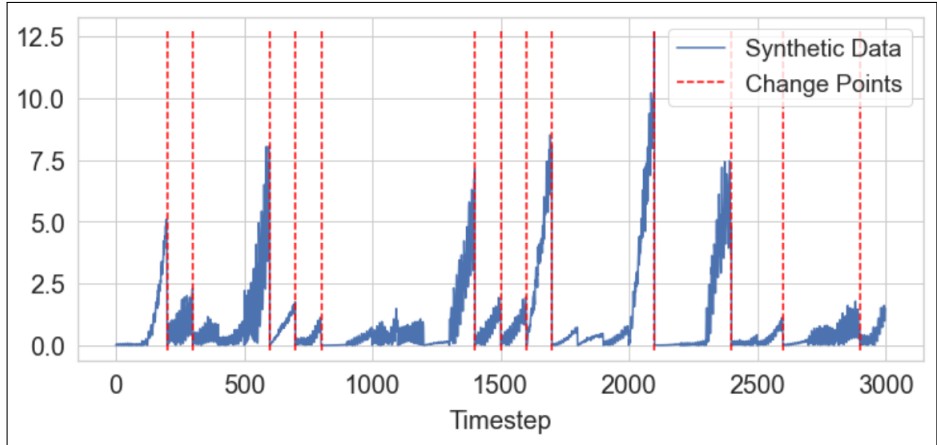

Figure 3: Synthetic data (blue) with manual change points (red) located at time indices 200, 300, 600, 700, 800, 1400, 1500, 1600, 1700, 2100, 2400, 2600, 2900.

Table 7: Prediction error (RMSE) of the different methods on the synthetic data. Reported are test and training RMSE of (I) naive, (II) DeepAR, and our BatchCP equipped with (III) manual and (IV) MOSUM change-point detection.

| SCENARIO | TRAIN | TEST |
|---|---|---|
| Baseline naive (I) | 3.57 | 9.33 |
| Transformer-No CPD (II) | 2.81 | 5.77 |
| Transformer-Manual (III) | 2.42 | 5.40 |
| Transformer-MOSUM (IV) | 2.51 | 5.62 |

## C    Treasury Rate (Univariate)

We now consider univariate data of the Federal Reserve Board One Year Treasury Rate Dataset. The samples are daily averages of yields of several treasury securities, all adjusted to the equivalent of a one-year maturity.

With our baseline modell (I) we get an RMSE of 3.98 for training samples and 8.17 for test samples.

The change points are are not immediately recognizable here. But we know that recessions have occurred at time points 1992, 3155, 4544, 5105, 7065, 9710, 11480, 14543—see Figure 6. MOSUM recognizes these change points relatively accurately again (for detailed MOSUM results of the treasury dataset see Appendix B). Algorithm 1 yields the maximal batch size

$$s_{max} = \left\lceil \frac{4544 - 3155}{2} \right\rceil = \left\lceil \frac{1389}{2} \right\rceil = 695\,.$$

We choose $s = 50$.

The results are summarized in Table 8. Once more, our BatchCP outmatches DeepAR: the test errors of BatchCP are about $9\,\%$ (Scenario (III)) and $5\,\%$ (Scenario (IV) better than for DeepAR and also much better than for naïve.

Figure 4 shows the treasury data with change points detected by MOSUM.

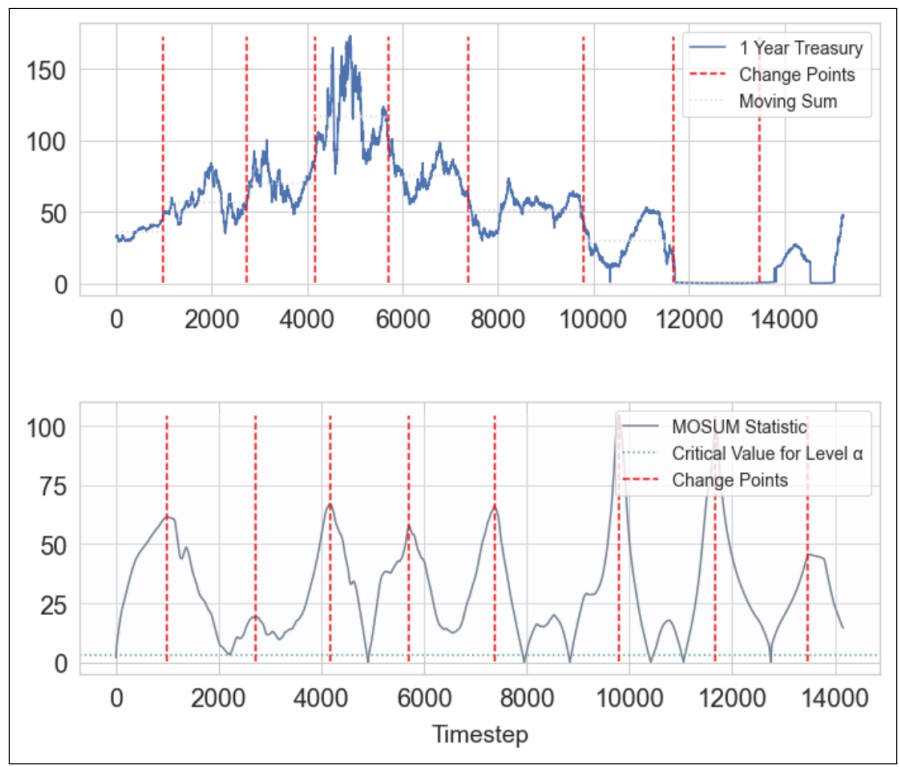

Figure 4: Upper diagram: The one-year-treasury time series (blue) with change points (red) detected with MOSUM, and the corresponding moving sum (grey). Lower diagram: The MOSUM statistic, which identifies the changes in the mean, and correspondingly the detected change points in red. The defined threshold value is plotted in grey.

Table 8: Prediction error (RMSE) of the different methods on the Treasury data. Reported are test and training RMSE of (I) naïve, (II) DeepAR, and our BatchCP equipped with (III) manual and (IV) MOSUM change-point detection.

| SCENARIO | TRAIN | TEST |
|---|---|---|
| Baseline naive (I) | 3.98 | 8.17 |
| Transformer-No CPD (II) | 4.13 | 5.93 |
| Transformer-Manual (III) | 3.40 | 5.37 |
| Transformer-MOSUM (IV) | 3.69 | 5.61 |

## D  Randomness of the Pipelines

The following boxplots that describe the results on the pollution data for five different seeds illustrate that our results do not hinge on specific seeds.

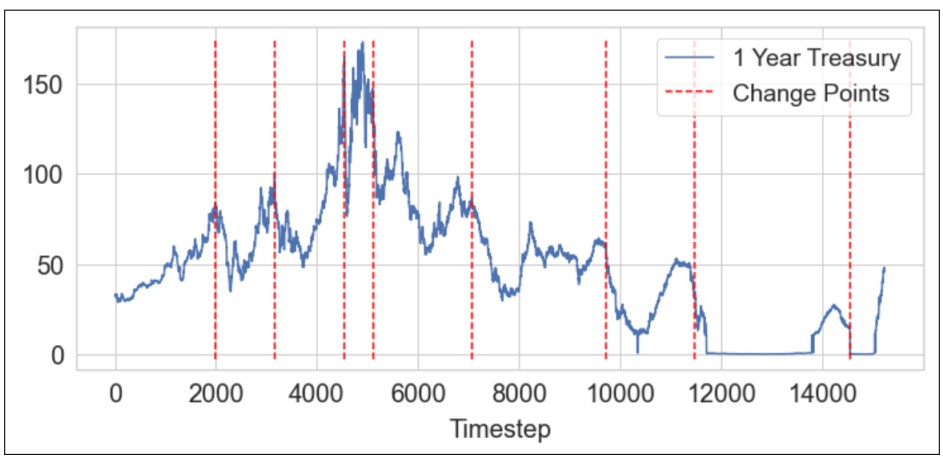

Figure 5: Treasury dataset (blue) with manual change points (red) located at time indices 1992, 3155, 4544, 5105, 7065, 9710, 11480, 14543.

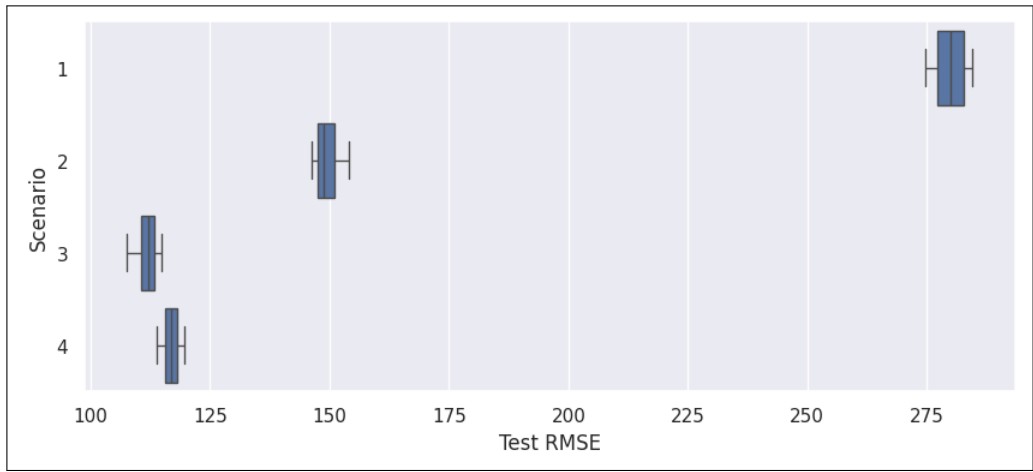

Figure 6: This diagram shows boxplots for scenarios 1 to 4 (Naive method, DeepAR-No CPD, BatchCP-Manual, BatchCP-MOSUM) with the seeds set at 10, 20, 30, 40, 50

