# OpenReview forum: "BatchCP: Forecasting Time-Series Data That Have Change Points"
_TMLR — Rejected by TMLR_

### Review · Reviewer_5ytP · 2025-08-06

**Summary Of Contributions:**

The proposed method is named as BatchCP, which is an algorithm for time series analysis that takes consideration of change points during batch generation. Specifically, given a change point detection method A, the algorithm uses A to detect change point, then define segments of the time series to compute the window length. Later, batches used during training are generated by removing the time segments that overlaps with detected change points. The proposed method is evaluated on pollution, football, electricity, and synthetic datasets, and is benchmarked with a naive forecasting method, a standard forecasting method (DeepAR), DeepAR + BatchCP with manually generated change points, and DeepAR + BatchCP with auto generated change points. The authors further experimented BatchCP + Transformers, and applying this on multivariate time series datasets.

Strength: Simple method that can be flexibly plugged into any training architecture.
Weakness:
1. Lack of novelty - How is the proposed method any different from the "Segmentation approaches" mentioned inside the introduction section, which "split time series at detected change points and treat each segment independently"?
2. Lack of robust baselines and organized expeirments - Overall, the experiments are poorly conducted.
- The authors did not compare the proposed method with state-of-the-art methods.
- The authors did not benchmark the proposed method on standard time series forecasting evaluation datasets.
- Datasets used are all small-scale.
- The authors did not include different experimental settings, e.g. different forecasting lengths.
3. Lack of ablation and additional analysis. e.g. interpretability
4. Lack of theoretical results.

**Audience:**

No

**Audience Explanation:**

The proposed method can easily be replaced with a segmentation method, combined with a sampler in PyTorch. It is unclear to me how the sampling algorithm by itself is useful, both experimentally and theoretically.

**Broader Impact Concerns:**

NA.

**Claims And Evidence:**

No

**Claims Explanation:**

The proposed method is only marginally different from the "segmentation method" that is mentioned in the introduction section. The authors did not directly compare the proposed method with the "segmentation method" mentioned inside the introduction that is highly similar to BatchCP. Aside from that, overall, the experiments are poorly designed and conducted, being significantly worse than the quality of standard forecasting research papers.

**Requested Changes:**

NA.

---

### Review · Reviewer_r1rp · 2025-08-19

**Summary Of Contributions:**

This paper studies time series forecasting with change points and proposes a framework named BatchCP. The key idea is to adaptively partition the series so that no batch contains change points. This avoids semantic confusion within batches and improves forecasting performance. Therefore, BatchCP can be applied to existing forecasting methods while keeping the original time series structure unchanged.

**Strengths:**
1. In my view, it is meaningful to address change points in a way that preserves the integrity of the original time series.
2. The method is simple and easy to understand.
3. Partial pseudo-code is provided, which helps in understanding the paper.

**Weaknesses:**
1. The motivation is unclear. The authors do not highlight why it is important to address change points and what negative impacts they have on time series forecasting.
2. The current experiments are insufficient to fully support the paper, specifically:
   a. Only a single Transformer or DeepAR is used as the backbone for each dataset, which is unconvincing.
   b. Only one metric (RMSE) is used for evaluation.
   c. No hyper-parameter experiments, especially BatchCP is highly dependent on batch size.
   d. No comparison is made with previous methods that focus on time series with change points. Although those methods modify the time series structure, in my view, they should still be compared. Moreover, since the Introduction Section already mentions many such methods, the lack of comparison is a clear gap.
   e. In my opinion, BatchCP needs to be combined with automated change point detection methods, since real-world datasets are large and cannot be handled manually. However, the paper only evaluates MOSUM.
   f. The experimental analysis is insufficient. For example, in Table 2, why does BatchCP+MOSUM perform worse than methods that ignore change points?
3. Writing and formatting need significant improvement:
   a. The introduction mainly consists of related work, without emphasizing the research significance. Related work should be placed in a separate section.
   b. Many paragraphs are overly lengthy.
   c. In Table 6, the baseline should be DeepVAR.
   d. Acronyms are not used appropriately, for example, some acronyms are introduced without providing their full terms (e.g., CPD).

**Audience:**

Yes

**Audience Explanation:**

BatchCP can be used to handle change points in time series forecasting, which I find interesting.

**Claims And Evidence:**

No

**Claims Explanation:**

In my view, the experiments in this paper do not sufficiently support the claims. Please refer to the previous Weakness for details.

**Requested Changes:**

Please refer to the previous Weakness for details.

---

> ### Author Response · Authors · 2025-09-05
>
> We sincerely thank you for the quick and comprehensive review. We would like to ask a few follow-up questions regarding the identified limitations:
> 1. What alternative approaches to change point detection would you recommend?
> 2. What additional backbone architectures would you suggest besides DeepAR and Transformer models?
>
> Thank you in advance and thank you for your efforts.

---

> ### Comment · Reviewer_r1rp · 2025-09-28
> **Reply**
>
> 1. I suggest trying [1] and [2].
> 2. You could use PatchTST [3], Informer [4], and etc.
>
> [1] Unsupervised Change Point Detection in Multivariate Time Series.
> [2] Modeling Piece-Wise Stationary Time Series.
> [3] A time series is worth 64 words: Long-term forecasting with transformers.
> [4] Informer: Beyond Efficient Transformer for Long Sequence Time-Series Forecasting.

---

### Review · Reviewer_BLU8 · 2025-09-01

**Summary Of Contributions:**

This paper proposes BatchCP, a training-time framework for deep time series forecasters that first detects change points and then forbids any training window that would span a change point. Concretely, the authors define segments $S_0,…,S_k$ induced by detected CPs, sample training windows only from within a single segment using the set of valid starts $W_{valid}$, and select the maximum window length by $L_{max} = ⌊min|S_i|/2⌋$ to guarantee that each segment contributes at least one window. The method is model agnostic and is applied without architectural changes to DeepAR, Transformer based models, and DeepVAR. Across several datasets (industrial pollution, football goal differences, electricity consumption, synthetic, and a treasury series), BatchCP improves RMSE over ‘vanilla’ training that ignores change points.

**Additional Comments:**

The related work section is timely but could be broadened to encompass emergent LLM‑based approaches to time‑series anomaly detection and CP reasoning. Recent studies use in‑context instructions to define CP/anomaly semantics with frozen LLMs, and multimodal LLMs that consume plotted time‑series images with explicit CP conditions; these lines of work would make useful contrasts because they handle CPs at inference time rather than only at training time [1, 2]. Positioning BatchCP relative to such detector‑assisted inference strategies would strengthen the paper’s contribution.

[1] Zhou et al., Can LLMs Understand Time Series Anomalies?, arXiv 2024

[2] Xu et al., Can Multimodal LLMs Perform Time Series Anomaly Detection?, arXiv 2025

**Audience:**

Yes

**Audience Explanation:**

Change points are pervasive across industrial, financial, and environmental time series, and BatchCP provides a simple, model agnostic mechanism for batch generation that practitioners can adopt without modifying architectures or loss functions.
The empirical improvements on several datasets and the clean integration into standard training loops would interest both researchers and applied teams who routinely face regime changes. The idea that a data loader policy alone can yield consistent gains is compelling and practically relevant.

**Broader Impact Concerns:**

If CPs are over-detected (e.g., because transient anomalies are mistaken for structural breaks), models may underutilize data and become brittle. Conversely, if CPs are missed, training may still be contaminated by cross-regime windows. Nevertheless, no ethical issues are apparent.

**Claims And Evidence:**

No

**Claims Explanation:**

The main empirical claim that ‘excluding training batches that cross change points improves forecasting accuracy’ is supported by multiple RMSE tables where BatchCP beats the vanilla baselines on DeepAR and Transformer variants, including the multivariate DeepVAR setting, and by a seed  robustness figure in the appendix. However, several central assertions remain either unclear or insufficiently evidenced.
First, the paper states in the Introduction that “uncertainty estimates become unreliable around historical change points,” but no uncertainty or calibration metrics (e.g., coverage, interval width, CRPS) are reported. This weakens the support for claims about uncertainty degradation and reliability.
Second, a definitional inconsistency arises between Section 2 and Figure 1. Equation 1 defines each change point $c_j$ as the first index of $S_j$, which implies that a window starting at $t=c_j$ and remaining within $S_j$ is valid. However, the authors later state that one must “ensure no batch $[i_{start}, i_{end}]$ contains any change point $c_j$,” while the caption of Figure 1 emphasizes that a window is permissible only if the change point (CP) lies outside the batch. These two statements conflict on whether a window may include the CP location itself, thereby creating ambiguity regarding the precise boundary. For instance, should the green permissible batch area on the right side of Figure 1 not include $t=7$ to be consistent with Equation 1 in Section 2?
Third, the method claims to “maximize information utilization” via $L_{max} = ⌊min|S_i|/2⌋$ to retain the “complete dataset”. However, this choice may be overly conservative and can result in short, leftover tails near CPs unused. The paper does not analyze the distribution of valid start positions $W_{valid}$, the acceptance ratio ρ, or how much data near CPs is actually excluded for typical datasets; nor does it provide theoretical lower bounds on ρ beyond informal discussion. Would there be a case where a valid batch size exists that is smaller than $min_{i=0,...,k}|S_i|$ but larger than $min_{i=0,...,k}|S_i|/2$, without the division by two? Would this not contradict the claim of "maximizing information utilization"?

Fourth, the “universal” claim would be stronger with broader architectural coverage and baselines. Although Transformers and DeepAR/DeepVAR are tested, TFT--explicitly cited as a target architecture--is not included in the experiments. Furthermore, no comparison is provided against segmentation based baselines (training per segment) or against robust/CP aware training alternatives.

Finally, some internal inconsistencies reduce clarity and weaken confidence. Algorithm 2 first computes $W_{valid}$ and then samples uniformly from it, rendering the subsequent “repeat…until valid window found” loop redundant. In addition, the synthetic data table in the appendix labels the baseline as “Transformer No CPD” while the accompanying text refers to DeepAR, creating confusion.

**Requested Changes:**

Please clarify the definition of a valid window at change point boundaries. Equation 1 and the paragraph that forbids any batch “containing” a CP, together with Figure 1’s captions, give conflicting rules. Please choose one convention and revise the equations, algorithms, and figure captions accordingly. If windows that start at $c_j$ are allowed (because they lie entirely in $S_j$ ), say so and update the captions. If not, redefine segments or $W_{valid}$ to exclude $c_j$ explicitly.
Please eliminate potential data leakage and clarify the split protocol. For all datasets and for both “manual” and “MOSUM” scenarios, is should be stated explicitly whether CP detection relies solely on training data and how CPs within validation/test periods are handled. If detection requires full series access, report results with train only detection to quantify any gap. Align Algorithm 1 with the experimental protocol in Section 4.
The term “information utilization” should be quantitatively defined. For each dataset, report the histogram of valid start positions $W_{valid}$, the fraction $ρ = |W_{valid}|/(T − L + 1)$, and how many samples are excluded near each CP. Provide theoretical guarantees or empirical bounds that $L ≤ ⌊min|S_i|/2⌋$ yields a practical lower bound on ρ, and discuss failure modes when change points are dense. Please include the statistics on the acceptance ratio statistics and wall clock overhead for batch generation.
Please address the leftover segment edge cases. For consecutive CPs separated by small gaps (e.g., lengths 4 and 5), $L_{max} = 2$ creates a leftover of length 1 that never appears in any window. It should be stated whether such tails are discarded or down weighted and this should be reconciled with the claim of “preserving the complete dataset.” Consider reporting results with alternative strategies such as variable window lengths per segment, selecting L as the largest value in $(min|S_i|/2, min|S_i|]$ that satisfies a minimum ρ, or curriculum over L.
Please evaluate uncertainty and calibration. Since the Introduction emphasizes unreliable uncertainty near CPs, add CRPS, interval coverage, and sharpness metrics, and analyze calibration around detected CP neighborhoods versus far from CPs. This will directly test the uncertainty claim, not just point accuracy.
Algorithm 2 and its notation should be clarified. If $W_{valid}$ is computed first and sampling is performed directly from $W_{valid}$, the “repeat…until valid window found” loop is unnecessary; otherwise, show the rejection sampling version explicitly. Also, the notation for window length/batch size (s vs. L) should be unified across the text, algorithms, and figure captions, and the table labels in the synthetic and treasury appendices should be double-checked for consistency.
Please differentiate change points from anomalies. Since MOSUM/CUSUM primarily detect mean shifts, discuss and quantify how often transient anomalies are labeled as CPs and how that affects BatchCP. It would be beneficial for the authors to report precision/recall for CP detection against the “manual” labels, to demonstrate sensitivity to false positives/negatives, and to include a stress test in which the detector is deliberately noisy.

---

> ### Author Response · Authors · 2025-09-05
>
> We sincerely thank you for the prompt and detailed report. We would like to ask a few follow-up questions regarding your points:
> 1. Which specific probabilistic baselines would you recommend for meaningful CRPS and calibration comparisons?
> 2. What type of theoretical bounds for the acceptance ratio would be most valuable from your perspective?
> 3. How should we concretely implement the train-only CP detection comparison - which detection methods would you recommend?
>
> Thank you in advance and once again, thank you for your efforts.

---

> > ### Comment · Reviewer_BLU8 · 2025-09-15
> > **Comment for Authors**
> >
> > - Regarding calibration, it would be sufficient to evaluate one or two widely used probabilistic forecasting models (e.g., DeepAR or TFT) and report calibration metrics such as CRPS and coverage.
> > - Regarding the acceptance ratio, even a simple theoretical lower bound in terms of the minimum segment length and the chosen window length $L$, together with empirical statistics (e.g., the distribution of valid window start positions, observed acceptance ratio $\rho$, and the proportion of excluded samples), would substantially strengthen the claim.
> > - Regarding train-only detection, presenting results with a standard detector such as MOSUM or CUSUM applied solely to the training split would help clarify the potential leakage issue.

---

### Decision · Action_Editor_kTmz · 2025-10-09

**Recommendation:** Reject

**Audience:**

No

**Audience Explanation:**

The authors did not submit a revision to address the reviewers' concerns.  Hence, the quality of the paper is likely not sufficient for TMLR's audience.

**Claims And Evidence:**

No

**Claims Explanation:**

The reviewers have concerns with the original submission.  However, the authors did not try to address the issues and submit a revision.   The concerns include the proposed method is similar to a segmentation method, limited comparison with existing methods, some claims are not well supported, some descriptions of the method are inconsistent, and lack of ablation studies to analyze the method.